# Assessment of Spatial and Vertical Variability of Water Quality: Case Study of a Polymictic Polish Lake

**DOI:** 10.3390/ijerph18168620

**Published:** 2021-08-15

**Authors:** Beata Ferencz, Jarosław Dawidek

**Affiliations:** 1Department of Hydrobiology and Protection of Ecosystems, Faculty of Environmental Biology, University of Life Sciences, 13 Akademicka St, 20-950 Lublin, Poland; 2Department of Hydrology and Climatology, Faculty of Earth Sciences and Spatial Management, Maria Curie-Skłodowska University, Kraśnicka 2cd, 20-718 Lublin, Poland; jaroslaw.dawidek@poczta.umcs.lublin.pl

**Keywords:** water mixing, inflow-outflow transect, dissolved oxygen, pH, chlorophyll a

## Abstract

UE regulations focus on methods of water quality monitoring and their use in rational management practices. This study investigated horizontal and vertical variations of electrical conductivity (EC), pH, dissolved oxygen (DO), and chlorophyll a (Chl-a) in a shallow polymictic lake. Monitoring of short-term variability of physical and chemical lake water parameters is a critical component in lake management, as it influences aquatic life. Based on the field research, maps of spatial distribution of the parameters were drawn. Using two methods: (1) a classical approach to water column measurements, from the top to the bottom (TB), in which the reference point is always a surface layer (SL), and (2) a newly introduced method of lake water quality monitoring based on a nearest neighbor (NN) approach; a comparison of higher and lower layers of the water column. By subtracting partial maps of spatial variability for different depths, final raster images were obtained. The NN method is rather absent in the limnology literature worldwide. Vertical and horizontal variability of the tested parameters in the polymictic, shallow Lake Bikcze (Poland) was presented in the results. In the presented paper, the commonly used TB method emphasized the role of the surface layer in shaping the variability of physicochemical parameters of lake waters. It shows a general trend of parameters’ changes from the top, to the bottom. The newly presented NN method, which has a major advantage in its simplicity and objectivity, emphasized structural differentiation within the range of variability. The nearest neighbor method was more accurate in showing the actual structure of fluctuation of parameters with higher fluctuation in the water column. Its advantage is a detailed recognition of the vertical variability of selected parameters in the water column. The method may be used regardless of the lake depth, its location in climatic zone, and/or region.

## 1. Introduction

Lakes are a very important source of water for drinking, irrigation, cooling, recreation, and [1], at the same time, are impacted by anthropogenic factors that modify both external (catchment) and internal (in-lake) originated processes [2]. Shallow lakes are widespread worldwide. They are often subject to conservation policies [3] because of their susceptibility to eutrophication [4]. Shallow lakes do not stratify, so the water column is frequently mixed [5]. Hence, they are highly dynamic systems [6]. The lakes are complex ecosystems strongly influenced by physical, chemical, and biological processes arising from sediment–water interaction and the potential impact of aquatic vegetation [7]. High instability of the water bodies cause the close link between bottom sediments and overlying waters [8]. The main environmental factors that determine aquatic life in the lake are parameters that include pH, dissolved oxygen, conductivity, or chlorophyll-a [9]. Deterioration of the quality of lake water, due to human impact, alters physicochemical parameters, influencing aquatic organisms [10].

Water quality assessment in lakes is a widely recognized problem [11,12]. For understanding water quality conditions, monitoring the physical–chemical parameters of lakes’ water is crucial, because it provides essential information for efficient water management practices. In general water quality can be investigated based on physical, chemical, or biological properties [13]. The interaction of the physical and chemical properties of water usually determine the composition, distribution, and abundance of aquatic life in lake’s ecosystem [14]. The biological characteristic of water quality is of paramount importance for human health [15]. Thus, various water quality indicators including physical, chemical, and biological properties are widely used in water quality monitoring [16,17,18]. Among them, statistical [19,20], mathematical [21,22], and remote sensing [23,24,25] are most popular.

The paper presents two methods for testing the variability of quality parameters in the lake’s water column: (1) classical from top to bottom (TB) approach, in which the reference point is always the surface layer (SL) and (2) newly presented nearest neighbor (NN) approach, comparing the values of higher and lower layers of the water column. The TB method dominates in limnological analyses, such as oxygen, thermal and chemical stratification [26,27,28]. The NN method is rather absent in the limnological literature. The main advantage of this method is detailed analysis of vertical variation of selected parameters in the water column. In contrast to the TB approach, each layer of the water column is equally important. For both methods to be effective, measurement data should be provided in real time. Furthermore, a use of automated sampling is required to increase sampling frequency under comparable weather conditions. The best solution would be using both presented methods simultaneously in lake monitoring studies. It would provide complete information of the variability of analyzed parameters. 

In a shallow polymictic lake, we hypothesized that homogeneous physicochemical conditions occur in the lake basin. The specific objectives of the study were: (a) assessment of spatial variability of water quality up to 1 m in depth; (b) identification of areas of significant parameter variation.

### Study Area

The measurements were conducted in the shallow, polymictic Lake Bikcze, during spring mixing time (20 May 2017) to ensure stable and comparable weather conditions during the measurements. The average daily air temperature (10.9 °C) represented spring conditions. The wind velocity from the SW direction was 3.7 m s^−1^. Air humidity was 77%, while atmospheric pressure was 1011 hPa.

Lake Bikcze is one of around 70 Łęczna-Włodawa lakes. These lakes are the only group of lakes in Poland, occurring beyond the boundary of the last glaciation. They are characterized by shallow basins (only 5 lakes are over 20 m deep) and low developed shorelines. The area of the largest lake, Lake Usciwierz, is only 275 ha. The deepest, Lake Piaseczno, is 39 m deep. Lake Bikcze has been included into Wieprz-Krzna Canal drainage system. The canal construction from 1954 to 1961 altered water distribution in the Łęczna-Włodawa Lake District area. The basin of Lake Bikcze was converted into a specific semi-anthropogenic reservoir in 1969. Lake shores were embanked, and a drainage ditch was constructed along the western lakeshore with its outlet on the north side to the deepened Piwonia River channel (Figure 1).

However, the lake is supplied by a surface tributary and drained by the outlet. The lake area is 759,038 m^2^, and the volume is 988,482 m^3^ (Table 1). 

The lake is one of the least-recognized reservoirs in the Łęczna-Włodawa Lake District. In the shallow lake under study, thermal stratification did not occur. Hydrometeorological conditions on the day of sampling favored water mixing. A prevalence of outlet discharge (14.8 L s^−1^) over inlet (13.7 L s^−1^) was observed. This indicated a domination of the drainage role, especially in the northern part of the lake basin. Water flushing time, calculated as the ratio of the lake volume to the outlet discharge, amounted to 780 days.

## 2. Materials and Methods

The lake was sampled in 72 profiles, evenly distributed within the lake basin. Geographical location of the sampling points was determined with global positioning system (GPS), under unconstructive conditions (clear sky). In the abovementioned profiles, at the surface level (SL) and at the depths of 0.5 and 1 m physicochemical parameters, electrical conductivity (EC), dissolved oxygen (DO), chlorophyll and pH were measured using a YSI 6600 V2-4 Multi-parameter Water Quality Sonde. Chlorophyll fluorescence measurements were compared with extracted chlorophyll a to calculate Chl-a concentration according to standard methods of chlorophyll quantification. Reference samples of chlorophyll a were analyzed by the acetone extraction method [29] and corrected for pheophytin. The long-term amplitude of the Lake Bikcze water levels amounted to 87 cm, which is why the 1 m layer was adopted as the zone of active water exchange in the lake. Measurements of vertical variability of the selected parameters were performed at 0.5 m interval.

For the subsurface layer and 0.5 and 1 m depths, spatial maps of parameter distribution were made. Inverse distance weight (IDW) interpolation method was used in order to generate parameter distribution maps [30]. ArcView 9.1 was used to generate the maps with a grid resolution of 7 × 7 m [31]. The IDW interpolation method is based on the principle of assigning higher weights to data points closest to an unvisited point relative to those that are further away. Thus, the assigned weight is a function of the inverse distance as represented in the following formula:(1)$$f\left(x,y\right)=\left[\overset{N}{\underset{i=1}{\sum }}w(d_{i})z_{i}\right]\;÷\;\left[\overset{N}{\underset{i=1}{\sum }}w\left(d_{i}\right)\right]$$
where *f*(*x*,*y*) is the interpolated value at point (*x*,*y*), *w*(*di*) is the weighting function, *z_i_* is the data value at point *i*, and *d_i_* is the distance from point (*x*,*y*). The interpolated values of all points within the dataset are bound by min (*z_i_*) < *f*(*x*,*y*) < max (*z_i_*), as long as *w*(*d_i_*) > 0 [32]. The IDW interpolation method has been widely used on many types of data [30], especially in spatial analysis of environmental factors [33,34,35,36,37]. A map calculator was used to estimate the difference between each parameter surface value and the value at a depth of 0.5 m, and the difference between SL and the value at 1 m depth. This way, both vertical and horizontal variability of selected parameters were obtained. In order to determine the variation of selected parameters, both horizontally and vertically, differential maps were prepared, using two methods.

First, the top to bottom (TB) approach, considered the fixed reference point that constituted the surface layer of the water column. Lower maps were subtracted each time from the surface reference value. The total result was therefore the recognition of the “classic” variability of the parameters studied. Using the following equations, the solution commonly used in limnology was obtained:(2)$$“0–0.5”=\frac{I_{0}-I_{0.5}}{I_{0}}$$
(3)$$“0–1”=\frac{I_{0}-I_{1}}{I_{o}}$$

Second, the nearest neighbor (NN) approach was used and the differences between neighboring layers were calculated. The upper layer was always a minuend one. In this case, the structure of relative variability was obtained, which indicated the vertical rate of change of parameters. The first step considered comparison of data for the surface layer (SL), and the depth 0.5 m, thus it was equal to “0–0.5”. The next step was a comparison of depths “0.5–1”: (4)$$“0.5–1”=\frac{I_{0.5}-I_{1}}{I_{0.5}}$$
where *I*_0_, *I*_0.5_, *I*_1_, isobath 0, 0.5, 1 m, respectively 

The values of TB and NN changes of the analyzed parameters could be positive or negative. The value “0” meant homogeneous conditions, positive values indicated a decrease in the value of the parameter in the lower layer, while its negative values suggested its increased depth. Both approaches were treated complementarily, in order to obtain greater interpretability.

In the next step, maps of mean values (MV) for both methods were made; (1) arithmetic average of maps showing “0–0.5” and “0–1”; (2) arithmetic average of maps presenting “0–0.5” and “0.5–1”, respectively.

Due to the a clear latitudinal orientation of the parameters variability presented in the resultant maps (both partial and average), in accordance with the lake’s inflow–outflow transect, the variability of the average differences of the parameters examined was calculated along this transect.

Correlation coefficients between measured physical–chemical parameters were also determined. Pearson correlations were calculated for the values from the maps of mean values (MV), every 50 m of distance, along the inflow–outflow transect.

## 3. Results

Results showed that each of the tested parameters were characterized by variability in both horizontal and vertical layout. On the day of sampling, homogenous thermal conditions were observed (Table 2). Table 2 also presents a range of other physical–chemical parameters.

A clear latitudinal variability in all parameters was observed. Contour lines of parameters’ variability are almost perpendicular to the lake inflow–outflow line. The pH variability showed a moderate variation ranging from −0.0273 to 0.038. In variant “0–0.5”, negative values (in the range −0.0273 to −0.0167) were observed in the northern half of the lake basin and near the estuary of the tributary to the lake. The highest positive values (0.0148–0.0253) occurred approximately 300 m from the mouth of the tributary.

Negative values (up to −0.023) occurred only in small fragments of the outlet (mouth of outflow) at the difference of the surface and a meter depth layer (“0–1”). The highest values (in the range of 0.0258–0.038) were observed in the zone 350–500 m from the mouth of the inflow. The difference between the 0.5 m and 1 m layers (“0.5–1”) showed the least ordered pattern and the lowest range. While the zones of the lowest values (interval −0.0003 to −0.0001) coincided with “0–1”, the maximum values (range 0.0002–0.0003) occurred in several zones in the southern and central parts of the lake basin (Figure 2).

EC showed less variability than pH from −0.0137 to 0.0209 µS cm^−1^. Taking into account the difference between the surface and the 1 m layer, both the lowest and the highest values were observed in “0–1”. Maps presenting differences between 0.5 m and 1 m depths “0.5–1” showed a clear direction of EC changeability, that is, from the highest (from 0.0134 to 0.0197 µS cm^−1^) near the inflow to the lowest (from −0.0119 to −0.0055 µS cm^−1^) near the outflow. Results of “0–0.5” showed a sinusoidal pattern, with alternatively increasing and decreasing values.

Oxygen showed greater variability than pH and EC, ranging from −0.589 to 0.776 mg L^−1^, both in approach “0.5–1”. Taking into account the difference between the surface layer and 0.5 m, the “0–0.5” image was characterized by an alternating pattern of higher and lower values. The lowest (range −0.431 to −0.0266 mg L^−1^) occurred in the north-eastern part of the lake basin, the highest (0.0228–0.0392 mg L^−1^) in four zones, both in the southern and north part of the lake. The other two approaches showed a more ordered values distribution. In both maps, “0–1” and “0.5–1”, the highest values (0.0479–0.0681 and 0.0502–0.0775 mg L^−1^, respectively) occurred in two fragments of the southern and northern peripheries of the lake basin. The zone of lowest values (−0.0328 to −0.0126 mg L^−1^) in map presented “0–1” was observed at a distance of 400–600 m from the mouth of the inflow. In approach “0.5–1” (−0.0589 to −0.162 mg L^−1^), its location was similar, but a much smaller range was observed.

In the case of chlorophyll a, regardless of the approach, a clear decrease in values from the water inflow zone (south-eastern part of the basin) toward the northern peripheries of the lake basin (near the outflow) was observed. The above regularity was not observed for any other parameter. The values showed very large variations (from −1.199 to 0.3832 µg L^−1^). The smallest range of values was observed when comparing of 0.5 m and 1 m “0.5–1” (from −0.627 to 0.2954 µg L^−1^).

Depending on the method of calculation used, the spatial distribution of the average values of the parameters showed differentiation. In the case of each of the parameters, spatial distribution of values in the lake basin was similar to latitudinal. EC and Chl-a were characterized by a tendency to decrease from the south to the north. The average values were low in the case of pH in the inflow and outflow mouth zones, while elevated values were observed in the zone 350–500 m north of the inflow (Figure 3). When the values within supply and drainage zones were elevated, a different distribution of values occurred in the case of oxygen, while the lowest was recorded in the zone 350–550 m north of the inflow mouth.

A significant statistical relationship between EC and Chl-a (r = 0.62, *p* < 0.05), and pH and Chl-a (r = 0.66, *p* < 0.05), was observed regardless of the reference point (Table 3).

The analysis of values obtained from partial maps by the TB and NN method indicated the zonation of the pH and DO parameters, in which the differences between the limnetic zone and the fluvial zone (inflow and drainage) were clearly visible (Figure 3). Fluvial conditions shaped dissolved oxygen concentration to about 500 m from the place of tributary inflow and 1100 m from the place of basin drainage. Water DO concentration in the central zone of the basin (from 500 to 1100 m) resulted from the limnetic processes. In the case of pH, a similar pattern of dependence was observed. The fluvial conditions shaped the pH in zones 350 m from inflow and from 550 m to outflow. Only in the narrow transect zone (350–550 m) were limnetic conditions determined the pH variability.

Chlorophyll a showed a clear duality in results obtained from partial maps by the TB and NN methods. The fluvial conditions related to the water input determined Chl-a concentration in only 350 m of the transect. In the remaining part, the majority of the lake basin, the parameter was shaped by limnetic processes. In the case of EC, the synchronism of the results obtained with both methods was observed (Figure 4).

The results obtained from the MV maps confirmed the above-described trends. In the case of DO, the average differences calculated with the TB and NN method on the inflow–outflow transect had positive values in most cases, inverse to the pH differences (Figure 5). The lowest variability was demonstrated by EC. Differences ranged from −0.0012 to 0.0016 µS cm^−1^. In turn, the highest variability was shown by differences in average chlorophyll, that is, from −0.09 to 0.04 µg L^−1^. The smallest variation in the results of average differences was found in pH and DO (Figure 5).

## 4. Discussion

The very even distribution of temperature, both horizontally and vertically, was observed. Stable thermal conditions at the time of sampling favored the homogeneous conditions of oxygen solubility in water [38]. Higher temperature accelerates reproduction of phytoplankton, therefore, homogenization of chlorophyll a was also favored during sampling [39]. However, the studied parameters showed both vertical and horizontal variation, despite the fact that Lake Bikcze is a polymictic reservoir, flow-through, and is susceptible to wind influence [40].

The TB method presented in the paper emphasized the role of the surface layer in shaping the variability of physicochemical parameters of lake waters. The relative method, however, emphasized structural differentiation within the range of variability. In the case of parameters with very low variability, like the EC in Lake Bikcze, results obtained using both methods were comparable. The NN method better reflected the actual variability structure of parameters with lower stability in the water column.

Generally, water quality is highly variable over time [41]. The results showed that each of the tested parameters was characterized by variability in both horizontal and vertical layout in Lake Bikcze. A stable water parameter (3 °C variation both vertically and horizontally) was the temperature that indicated well mixed lake waters. Nevertheless, a clear latitudinal variation of all selected parameters was observed, almost perpendicular to the inflow and outflow line of the lake. This testified to the high role of streams (both inflow and outflow) in shaping selected quality parameters, superior to the wind direction (SW). Among the observed parameters, Chl-a showed the highest variability.

A significant statistical relationship between EC and Chl-a (r = 0.62, *p* < 0.05), and pH and Chl a (r = 0.66, *p* < 0.05) was observed, regardless of the reference point. A positive relationship between chlorophyll a and pH is usually explained by photosynthesis, because CO_2_ assimilation increases pH value [42]. Positive correlation of Chl-a and EC (r = 0.85, *p* < 0.05), considered as a surrogate for the concentration of nutrients in the water, has been previously observed in other areas [43]. However, in terms of Polish Lowlands, EC is mainly determined by bicarbonates. Although shallow lakes, especially polymictic ones, are exposed to wind and hence have a uniform distribution of physicochemical parameters in the water column, it has not been confirmed in the polymictic Lake Bikcze. The negative correlation between Chl-a and DO indicated that the exchange with the atmosphere was the main factor shaping DO [44]. Dissolved oxygen also showed high variability in the zone of inflow and outflow mouth, due to the impact of both, water movement, and atmosphere gas exchange [45]. A high value of DO, due to streams’ feeding has been previously observed by Romanescu and Stoleriu [46] in a temperate lake. Significant role of hydrological factors, in shaping DO concentration has also been previously established in Lake Bikcze. Correlation coefficients, calculated from the average maps, indicated higher values when using the relative method. This indicates that the relative method better reflects the actual variability of water quality parameters in the water column. The advantage of using the NN method is the ability to precisely identify the discontinuity zone between the part of the lake basin under the influence of fluvial processes and the one shaped by limnetic conditions and water drainage.

The highest variability of chlorophyll a suggested that processes shaping its concentration in the lake waters are complex and multidirectional [47]. Chl-a in Lake Bikcze is determined by in-lake processes rather than hydrological ones [40]. In the southern part of the lake basin, positive values of Chl-a differences indicated a higher concentration of the parameter in the upper layers of the water column. This was probably related to the supply of nutrients from the lake catchment with inflow waters [48]. Lakes often play a sink role, accumulating dissolved solids [49]. Therefore, elevated ion concentration can be observed in the part of the basin influenced by the inflow. In Lake Bikcze, an increased concentration of chlorophyll in the inflow zone may result from the agricultural nature of the inflow catchment area [50]. Reports from other areas show a contrary pattern, with deterioration of water quality from upstream to downstream of the lake basin [51]. Negative and constantly increasing values from about 350 m from the inflow’s mouths indicated a change of conditions from fluvial to limnetic ones. An increase in chlorophyll a concentration was observed in the deeper layers, which is typical of temperate lakes [52,53]. The abovementioned zone of the lake basin (about 350 m along the transect) was characterized by variability (both in plus and in minus) of other tested parameters (DO and pH). This confirmed a rapid change in conditions from fluvial to limnetic.

## 5. Conclusions

Freshwater monitoring is a key factor in sustainable water management policy. Shallow polymictic lakes are especially exposed to deterioration of water quality. The lakes lacking thermal stratification are known for homogeneous physical–chemical water parameters, resulting from continuous mixing. However, previous studies suggest that even in those well mixed water bodies, a spatial variability of water quality may be observed, both horizontally and vertically. Diurnal fluctuation in lake water quality parameters in lakes has already been investigated. However, direct observations of daily dynamics, in both the horizontal and vertical direction, are very rare. It is probably due to the fact than diurnal changes of weather conditions (e.g., wind velocity and/or air temperature), influence both physical and chemical water quality parameters. This paper presents two methods of establishing variability of most commonly measured parameters. In the newly presented NN approach, water quality parameters recorded in each depth in the water column is equally important. This is in contrast to the classical approach, in which each depth is referenced to the surface layer. The results show that the TB method, commonly used in limnology, emphasized the role of the surface layer in shaping the variability of physicochemical parameters of lake waters. Therefore, the from the top to the bottom method shows the general trend of changes in parameters from the surface to the assumed depth. The nearest neighbor method, however, is a better fit in terms of polymictic lake investigation. Within the typical variation shown by the TB method, there are zones (layers) in which anomalies in water quality parameters are observed, both in minus and in plus. Only the application of a NN method allows the identification of such zones and the interpretation of their causes.

## Figures and Tables

**Figure 1 ijerph-18-08620-f001:**
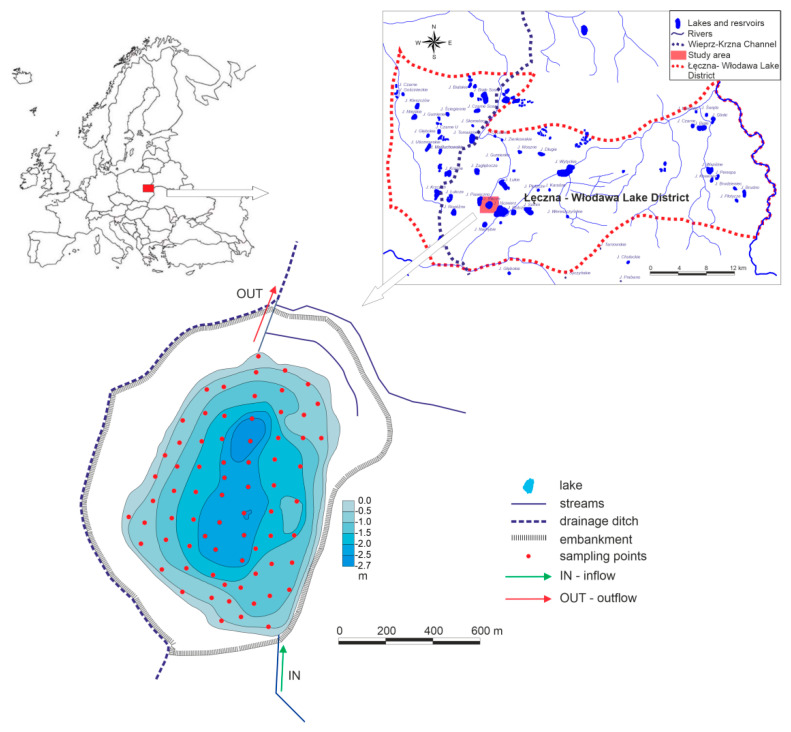
Location and bathymetric map of a study lake.

**Figure 2 ijerph-18-08620-f002:**
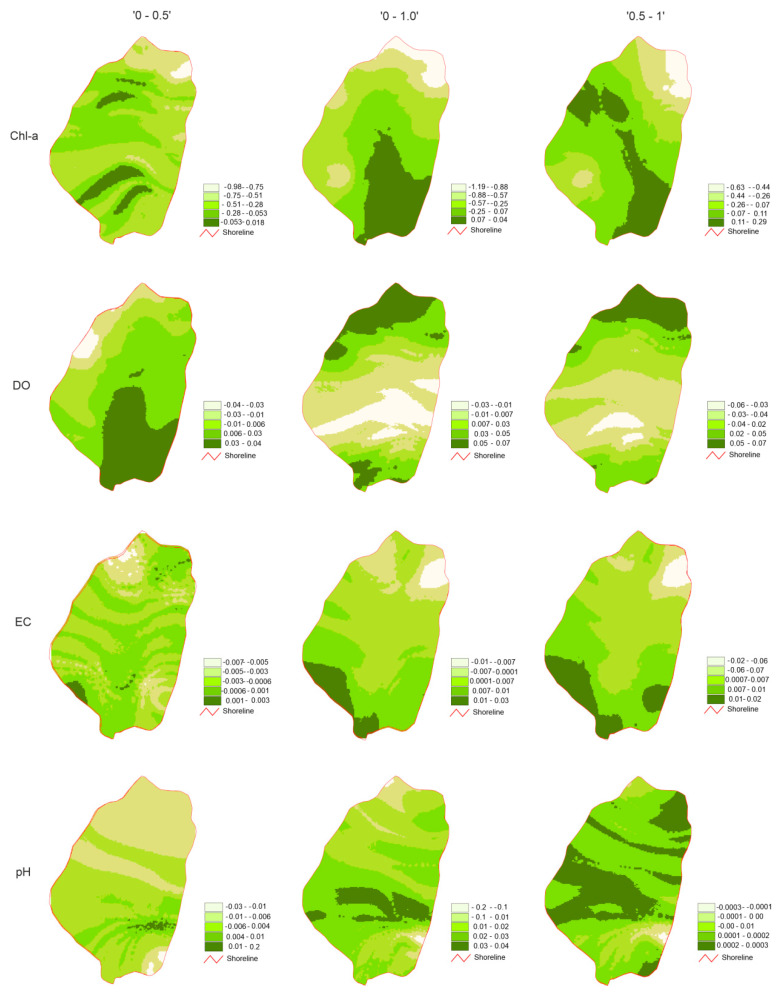
Partial differential maps calculated with TB and NN methods: “0–0.5”, “0–1”, “0.5–1”.

**Figure 3 ijerph-18-08620-f003:**
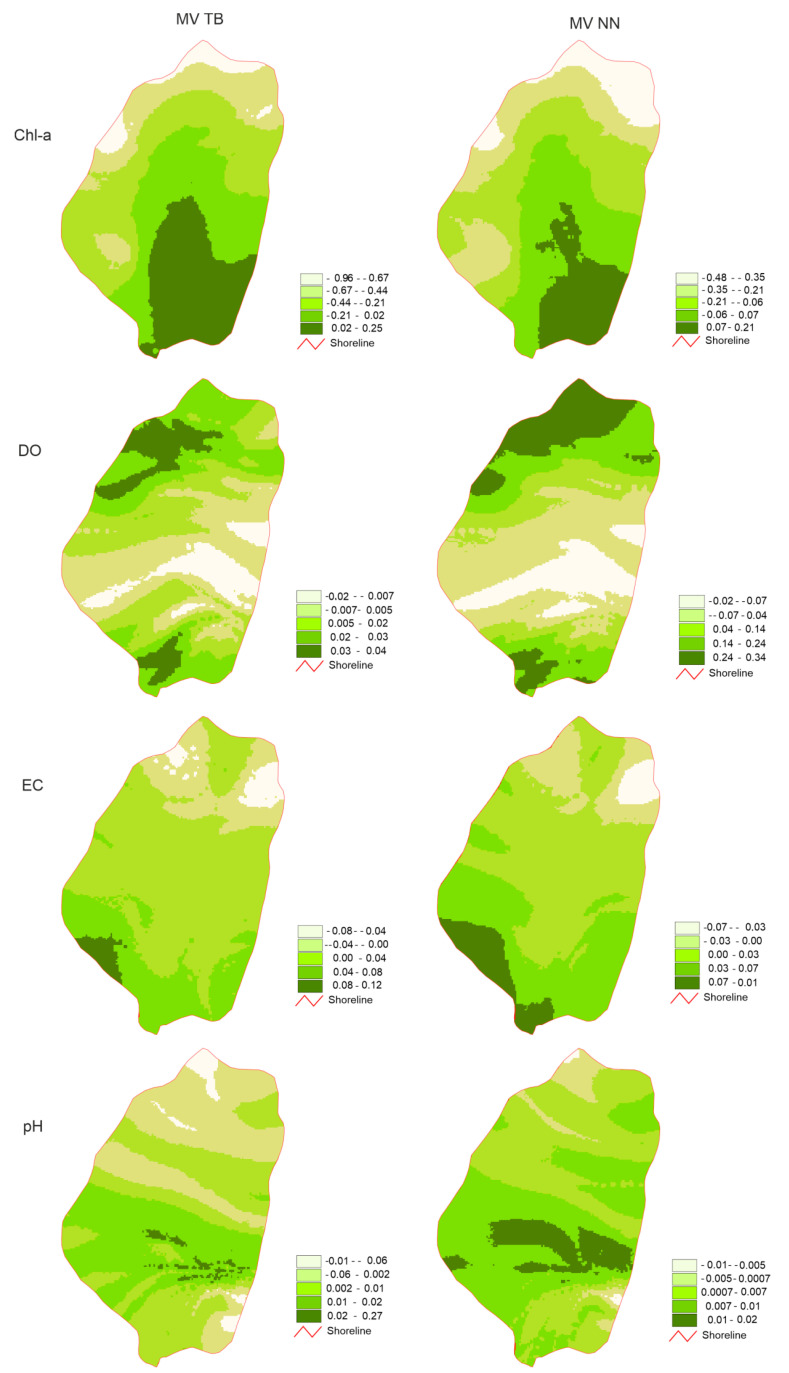
Maps of mean values (MV) for both TB (1) and NN (2) approaches.

**Figure 4 ijerph-18-08620-f004:**
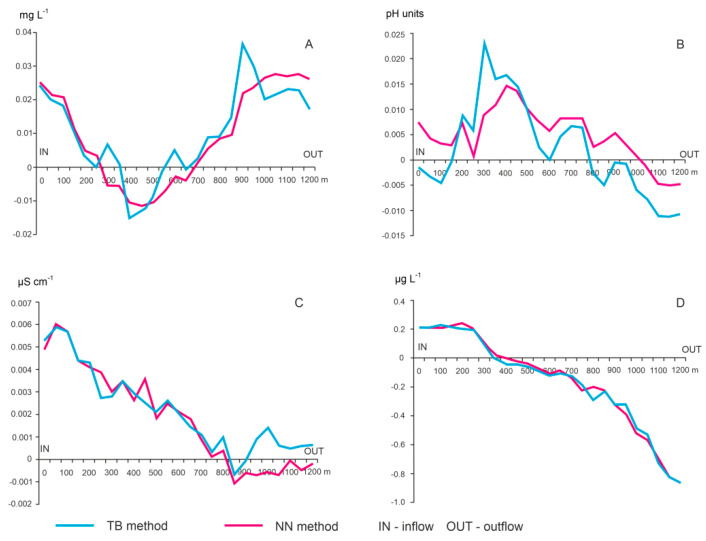
Variability of the parameters along lake’s inflow–outflow transect: (**A**) DO, (**B**) Ph, (**C**) EC, (**D**) Chl-a.

**Figure 5 ijerph-18-08620-f005:**
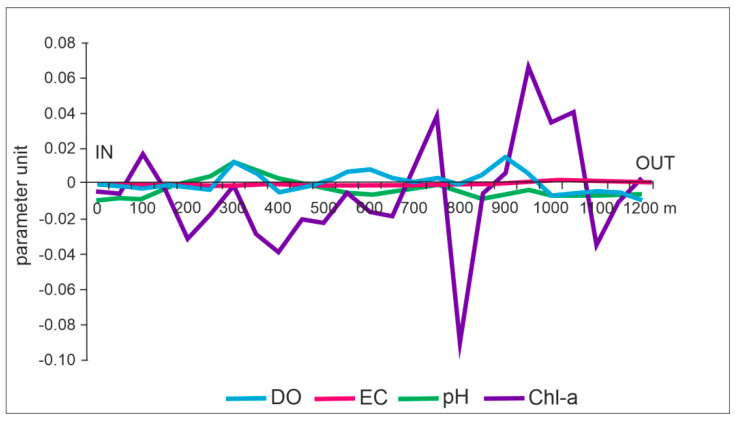
Differences (based on MV maps) between TB and NN values calculated for the selected parameters.

**Table 1 ijerph-18-08620-t001:** Morphometry of Lake Bikcze.

Parameter	Value	Unit
Area	759,038	m^2^
Volume	988,482	m^3^
Maximum depth	2.7	m
Mean depth	1.3	m
Maximum length	1206	m
Maximum width	860	m
Shoreline length	3347	m
Epilimnion depth	2.7	m

**Table 2 ijerph-18-08620-t002:** Range of values of selected water quality parameters (min–max ± standard deviation).

Depth	T °C	EC µS cm^−1^	pH	Chl a µg L^−1^	DO mg L^−1^
SL	15.49−16.6 ± 0.33	237−247 ± 2.91	7.1−8.32 ± 0.37	4−7.2 ± 0.75	8.34−10.95 ± 0.69
0.5	15.51−16-61 ± 0.33	237−247 ± 2.87	6.92−8.33 ± 0.39	3.2−5.4 ± 0.34	8.3−11.08 ±0.74
1.0	15.79−16.63 ± 0.31	238−245 ± 1.72	7.14−8.22 ± 0.32	3.8−6.6 ± 0.57	8.25−10.2 ± 0.54
1.5	15.51−16.61 ± 0.36	241−247 ± 2.49	6.92−7.89 ± 0.32	4−5.4 ± 0.22	8.3−9.31 ± 0.29
2.0	16.24−16.41 ± 0.08	241−242 ± 0.47	7.37−8.01 ± 0.27	5.0−5.8 ± 0.14	8.91−9.23 ± 0.14
2.5	16.23−16.38 ± 0.07	241−242 ± 0.47	7.36−8 ± 0.27	5.0−8.2 ± 0.68	8.29−9.27 ± 0.42

**Table 3 ijerph-18-08620-t003:** Correlation of variability of selected quality parameters for average values calculated for both TB and NN variants, along the transect inflow–outflow.

Parameter	DO	EC	pH	Chl-a
**TB**				
**DO**	1			
**EC**	−0.11. *p* =0.01	1		
**pH**	−0.71. *p* =0.06	0.13. *p* = 0.73	1	
**Chl-a**	−0.40. *p* = 0.02	0.62. *p* = 0.02	0.59. *p* = 0.02	1
**NN**				
**DO**	1			
**EC**	−0.31. *p* = 0.02	1		
**pH**	−0.81. *p* =0.25	0.43. *p* = 0.01	1	
**Chl-a**	−0.52. *p* =0.02	0.85. *p* = 0.03	0.66. *p* = 0.02	1

## Data Availability

The data that support the findings of this study are available on request from the corresponding author, Beata Ferencz.

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
