# Peer review of "Assessment of Spatial and Vertical Variability of Water Quality: Case Study of a Polymictic Polish Lake"

_ijerph, 2021, doi:10.3390/ijerph18168620_

Round 1
Reviewer 1 Report
I would like to suggest a proper statistical analysis to detect significant differences in the variations of the studied parameters horizontally and vertically in the shallow polymythic lake OR an explanation for not having performed it to analyze the results.
Author Response
Answers for Reviewer 1
I would like to suggest a proper statistical analysis to detect significant differences in the variations of the studied parameters horizontally and vertically in the shallow polymythic lake OR an explanation for not having performed it to analyze the results.
In the paper ‘Spatial and vertical variability of water quality in a polymictic Polish lake’ a graphic presentation was used, instead of ANOVA for the analysis of horizontal and vertical variability of the examined parameters. The advantage of this approach is the presentation of both the spatial variability and the vertical variability. Statistical analysis, regardless of the method, will show the strength of the relationship or the significance of the differences between the points, but it will not be possible to quickly locate those points (or a set of points - areas) affected by these differences. The graphical method allows to precisely indicate areas of anomalous differences, which increases the possibilities of interpreting the results.
Reviewer 2 Report
This manuscript seeks to compare the application of "absolute" and "relative" measurements of common limnological variables for the purposes of assessing lake water quality and/or status. If this is not the author's intended purpose, then I apologize for my misunderstanding. However there are numerous places where the manuscript does not do a good job of identifying its terms or goals. I think the manuscript requires major re-writing to assess whether or not it is actually acceptable for publication. In its current form, I cannot understand it well enough to make the determination.
- Manuscript needs grammatical editing. There are several places where there is a lack of singular/plural agreement between sentence sections (e.g. Lines 10-11).
- Line 29: remove the phrase "dilution of pollutants". This is not behavior that should be encouraged.
- Line 41: I don't think water quality assessment is an "issue". Choose another word.
- Within the Introduction, pay attention to the order of topics in sentences. Often times the final sentences should actually belong to the beginning of the paragraphs. This will ensure that all topics have the appropriate hierarchy within their paragraph. I often have to do the same re-writing with my own work, so I recognize it in others.
- Strongly reconsider the names "absolute" and "relative" for these two analyses. Given they are often paired with other descriptors such as "detailed" and "variation", reading comparisons of the two becomes confusing very quickly. Relative has an implication of being less precise in measurement, but that is not what is ultimately described.
- Were 72 points really measured in one day!? That seems like a lot. How did you account for differences made between measurements made at different times of day (e.g. 7am vs 7pm)? Some acknowledgement of this issue and how you dealt with it is necessary.
- Is this a series of pothole lakes? I think using that common limnological term would go a long way to placing these lakes in familiar context for many readers.
- Figure 1: Your scale needs units. Some of the elements in the key occur twice (e.g. streams), what is the difference? Please indicate which direction the lake flows. Since this comes up in your analysis a bit, it would be helpful to have the direction indicated on the map. Also, if the lake map part could be larger overall that would be helpful. This might help you fit all 72 points on the map (I counted and did not make it to 72).
- Table 1: Make sure the units in the text match the units in the table (e.g. area).
- Line 109: Term "dt" is not defined in the below text. Further more, there are some typographic formatting issues in the definitions.
- Within this section overall, there is a lot of confusing terminology. It is very difficult to follow if you are not well versed in this practice. From the text here and later discussions of it, it seems as though the terms IDW/Variant/Solution are used interchangeably once you get to the results here. They are painstakingly defined in the methods, but in the results, it's all over the place.
- Line 124: What is variable "I"? It is not defined. Also, you have the equations numbered, but do not refer to those numbers in the text, instead referring to "solution 1a". This is very confusing.
- If "solution 2a" is identical to "solution 1a", then why have 2a at all?
- Line 138: Why not have "2a" in the variant 2 formula? See above.
- Line 143: You only did the one set of correlations? For all the calculations, I would think an analysis directly comparing relative vs. absolute would be needed.
- Line 145: Figures 2 and 3 need to be mentioned before you get to 4 and 5.
- Line 154: Here is where we start to see the confusion of all the different terms. It is not clear what exactly these numbers are. Are these the result of "solution 1b"? "variant 2"? Differences in SD? What does it mean that there is "variation in the variant"? How does this range exemplify that? Additionally, you now refer to "variant 1a"...is this synonymous with "solution 1a"? How is the reader to know that?
- Table 2: The first column is untitled and has no units. I surmise it is depth, but as the reader I don't like making assumptions.
- Line 178: Again, we have "variant 2b" which I don't know how to distinguish from "solution 2b".
- Figure 2: Can you please label the solutions 1a, 1b, 2b as absolute or relative? Are there 2 different absolute measurements?
- Figure 3: How is this REALLY different than what is in Figure 2? How did you get from Fig 2 to Fig 3?
- Table 3: I get what's going on in this table, but your methods made it sound like you were going to be comparing position along the lake to each parameter. Alternatively, I think an overall analysis of parameter to parameter (regardless of transect position) would also be informative.
- Figure 4: The x-axis needs to be better labeled here somehow.
- Also, at what point will the measurements at various depths be discussed? You measured down to 3m in depth in places, yet we never see that data anywhere in this comparison. How do these other depths work into the absolute/relative calculation? Or the maps? This is not obvious.
- Figure 5. Is this 1a absolute or 1b absolute? Or the mean of absolute?
- Line 243: What was the wind on the day you measured? Had the lake just been mixed?
Author Response
Answers for Reviewer 2
This manuscript seeks to compare the application of "absolute" and "relative" measurements of common limnological variables for the purposes of assessing lake water quality and/or status. If this is not the author's intended purpose, then I apologize for my misunderstanding. However there are numerous places where the manuscript does not do a good job of identifying its terms or goals. I think the manuscript requires major re-writing to assess whether or not it is actually acceptable for publication. In its current form, I cannot understand it well enough to make the determination.
- Manuscript needs grammatical editing. There are several places where there is a lack of singular/plural agreement between sentence sections (e.g. Lines 10-11).
Grammatical editing of the paper was performed, according to the Reviewers suggestion. The manuscript has been professionally proofread by PRS (Proof-Reading-Service.com from UK).
- Line 29: remove the phrase "dilution of pollutants". This is not behavior that should be encouraged.
The phrase ‘dilution of pollutants’ has been removed, according to the Reviewers suggestion.
- Line 41: I don't think water quality assessment is an "issue". Choose another word.
The word ‘issue’ has been replaced by problem
- Within the Introduction, pay attention to the order of topics in sentences. Often times the final sentences should actually belong to the beginning of the paragraphs. This will ensure that all topics have the appropriate hierarchy within their paragraph. I often have to do the same re-writing with my own work, so I recognize it in others.
Introduction section has been modified in order to appropriate hierarchisation of the presented topics.
- Strongly reconsider the names "absolute" and "relative" for these two analyses. Given they are often paired with other descriptors such as "detailed" and "variation", reading comparisons of the two becomes confusing very quickly. Relative has an implication of being less precise in measurement, but that is not what is ultimately described.
According to the Relievers suggestion, the names ‘absolute’ and ‘relative’ have been changed. We agree that other names, such as From Top to bottom (TB) and Nearest neighbor (NN) are more self-descriptive.
- Were 72 points really measured in one day!? That seems like a lot. How did you account for differences made between measurements made at different times of day (e.g. 7am vs 7pm)? Some acknowledgement of this issue and how you dealt with it is necessary.
All the measurements were performed in one day. The atmospheric conditions that day were very homogeneous, (as we state in lines 85-88) which information is now stressed in the study area section. Temperature that day varied no more than 0.5oC. We have moved the part describing meteorological conditions of the day of sapling to the beginning of the paragraph, to clear this issue.
- Is this a series of pothole lakes? I think using that common limnological term would go a long way to placing these lakes in familiar context for many readers.
The lake is not a pothole lake. Łęczna-Włodawa lakes, as we described in the study area section, are located beyond the boundary of the last glaciation. Some of Masurian and Pomeranian lakes, formed during last glaciation are pothole lakes. What is more, the lake has been transformed into a specific semi-anthropogenic reservoir, as we described in line 71.
- Figure 1: Your scale needs units. Some of the elements in the key occur twice (e.g. streams), what is the difference? Please indicate which direction the lake flows. Since this comes up in your analysis a bit, it would be helpful to have the direction indicated on the map. Also, if the lake map part could be larger overall that would be helpful. This might help you fit all 72 points on the map (I counted and did not make it to 72).
The Reviewer was correct that Figure 1 needs edition. There should not be some elements in the legend, such as streams. In has been corrected. We also added units to the scale, indicated the direction of water flow, and enlarged the part of figure presenting the lake basin.
- Table 1: Make sure the units in the text match the units in the table (e.g. area).
Units in Table 1 have been change to match those in the text.
- Line 109: Term "dt" is not defined in the below text. Further more, there are some typographic formatting issues in the definitions.
Text has been reviewed and supplemented with explanations.
- Within this section overall, there is a lot of confusing terminology. It is very difficult to follow if you are not well versed in this practice. From the text here and later discussions of it, it seems as though the terms IDW/Variant/Solution are used interchangeably once you get to the results here. They are painstakingly defined in the methods, but in the results, it's all over the place.
Thank you for that remark. There was in fact a lot of confusing terminology in this section. We edited the text in order to clarify it. There should be no confusion now.
- Line 124: What is variable "I"? It is not defined. Also, you have the equations numbered, but do not refer to those numbers in the text, instead referring to "solution 1a". This is very confusing.
I is isobath, which is now clarified in the text. The ‘solutions’ were replaced by more clear abbreviation.
- If "solution 2a" is identical to "solution 1a", then why have 2a at all?
The confusing terminology was clarified
- Line 138: Why not have "2a" in the variant 2 formula? See above.
The confusing terminology was clarified
- Line 143: You only did the one set of correlations? For all the calculations, I would think an analysis directly comparing relative vs. absolute would be needed.
An analysis comparing the methods: from top to bottom and nearest neighbor, former relative and absolute methods, was performed on the inlet-outflow transect and the results are presented in figure 4. This solution allows to distinguish areas with the dominant process (river inflow, drainage or in-lake processes) shaping the analyzed parameters. For interpretation purpose, this is a better solution than the overall correlation. Therefore, only the correlations for individual parameters were analyzed.
- Line 145: Figures 2 and 3 need to be mentioned before you get to 4 and 5.
The figures 2 and 3 are now mentioned before 4 and 5
- Line 154: Here is where we start to see the confusion of all the different terms. It is not clear what exactly these numbers are. Are these the result of "solution 1b"? "variant 2"? Differences in SD? What does it mean that there is "variation in the variant"? How does this range exemplify that? Additionally, you now refer to "variant 1a"...is this synonymous with "solution 1a"? How is the reader to know that?
The confusing terminology was clarified
- Table 2: The first column is untitled and has no units. I surmise it is depth, but as the reader I don't like making assumptions.
The Reviewer is correct. It is depth. The Table has been corrected.
- Line 178: Again, we have "variant 2b" which I don't know how to distinguish from "solution 2b".
The confusing terminology was clarified
- Figure 2: Can you please label the solutions 1a, 1b, 2b as absolute or relative? Are there 2 different absolute measurements?
Labels have been change in order to clarify the Figure.
- Figure 3: How is this REALLY different than what is in Figure 2? How did you get from Fig 2 to Fig 3?
Additional information have been presented in the method section. Figure 3 presents mean values of maps presented in figure 2.
- Table 3: I get what's going on in this table, but your methods made it sound like you were going to be comparing position along the lake to each parameter. Alternatively, I think an overall analysis of parameter to parameter (regardless of transect position) would also be informative.
An overall analysis of parameter to parameter (regardless of transect position) would definitely be informative. However, we decided to limit the analysis to the transect, given the importance of the confluence processes in small lake basins. The processes of water exchange in polymictic lakes, are enhanced by the inflow and drainage of the lake basins. We aimed to highlight it in the paper.
- Figure 4: The x-axis needs to be better labeled here somehow.
The x-axix is better labeled now.
- Also, at what point will the measurements at various depths be discussed? You measured down to 3m in depth in places, yet we never see that data anywhere in this comparison. How do these other depths work into the absolute/relative calculation? Or the maps? This is not obvious.
The data were obtained up to a depth of 2.5 m, and they are presented in the table 2. Both the presented methods may be applied to any depth. The presentation of the results was limited to a depth of 1 m due to:· As we mentioned in line 98-101, the zone of active water exchange in Lake Bikcze is less than 1 m.· The shape of the lake basin, presented in Figure 1, indicates that while the number of measuring points in the analyzed layers is high, at a depth of 2.5 meters, it significantly decreases.
- Figure 5. Is this 1a absolute or 1b absolute? Or the mean of absolute?
The figure presents mean value results. It has been corrected.
- Line 243: What was the wind on the day you measured? Had the lake just been mixed?
All the meteorological conditions have been described in materials and methods section:
‘The average daily air temperature (10.9 °C) represented spring conditions. The wind velocity from the SW direction was 3.7 m s-1. Air humidity was 77%, while atmospheric pressure was 1011 hPa’.

Reviewer 3 Report
- The manuscript presents spatial and vertical variability of water quality in a polymictic Polish lake, which is interesting. The subject addressed is within the scope of the journal.
- However, the manuscript, in its present form, contains several weaknesses. Appropriate revisions to the following points should be undertaken in order to justify recommendation for publication.
- Full names should be shown for all abbreviations in their first occurrence in texts. For example, GPS in p.3, SD in p.5, etc.
- For readers to quickly catch your contribution, it would be better to highlight major difficulties and challenges, and your original achievements to overcome them, in a clearer way in abstract and introduction.
- It is shown in the reference list that the authors have a pertinent publication in this field. This raises some concerns regarding the potential overlap with their previous works. The authors should explicitly state the novel contribution of this work, the similarities, and the differences of this work with their previous publications.
- 1 - electrical conductivity, pH, dissolved oxygen, and chlorophyll a are adopted in the analysis. What are the other feasible alternatives? What are the advantages of adopting these parameters over others in this case? How will this affect the results? More details should be furnished.
- 1 - two approaches are adopted to obtain final raster images. What are the other feasible alternatives? What are the advantages of adopting these approaches over others in this case? How will this affect the results? More details should be furnished.
- 2 - a hypothesis that homogeneous physicochemical conditions occur in the lake basin is adopted in this study. What are other feasible alternatives? What are the advantages of adopting this hypothesis over others in this case? How will this affect the results? The authors should provide more details on this.
- 2 - Lake Bikcze is adopted as the case study. What are other feasible alternatives? What are the advantages of adopting this case study over others in this case? How will this affect the results? The authors should provide more details on this.
- 2 - historical records of 2017 are taken. Why are more recent data not included in the study? Is there any difficulty in obtaining more recent data? Are there any changes to situation in recent years? What are its effects on the result?
- 3 - inverse distance weight interpolation method is adopted to generate parameter distribution maps. What are other feasible alternatives? What are the advantages of adopting this method over others in this case? How will this affect the results? The authors should provide more details on this.
- 4 - Pearson correlation is adopted for the values from the maps. What are other feasible alternatives? What are the advantages of adopting this approach over others in this case? How will this affect the results? The authors should provide more details on this.
- 8-9 - “…In turn, the highest variability was shown by differences in average chlorophyll, that is, from −0.09 to 0.04 μg L-1. The smallest variation in the results of average differences was found in pH and DO.…” Some justification should be furnished on this issue.
- 10 - “…In the southern part of the lake basin, positive values of chl-a differences indicated a higher concentration of the parameter in the upper layers of the water column. This was probably related to the.…” More justification should be furnished on this issue.
- Some assumptions are stated in various sections. Justifications should be provided on these assumptions. Evaluation on how they will affect the results should be made.
- The discussion section in the present form is relatively weak and should be strengthened with more details and justifications.
- Moreover, the manuscript could be substantially improved by relying and citing more on recent literatures about contemporary real-life water quality case studies such as the followings. Discussions about result comparison and/or incorporation of those concepts in your works are encouraged:
- Setshedi, K.J., et al., “The Use of Artificial Neural Networks to Predict the Physicochemical Characteristics of Water Quality in Three District Municipalities, Eastern Cape Province, South Africa,” International Journal of Environmental Research and Public Health 18 (10): 5248 2021.
- Deng, T.A., et al., “Machine Learning Based Marine Water Quality Prediction for Coastal Hydro-environment Management,” Journal of Environmental Management 284: 112051 2021.
- Wang, B.B., et al., “Improved water pollution index for determining spatiotemporal water quality dynamics: Case study in the Erdao Songhua River Basin, China,” Ecological Indicators 129: 107931 2021.
- Some inconsistencies and minor errors that needed attention are:
- Replace “…were to a) asses spatial variability…” with “…were to a) assess spatial variability…” in line 60 of p.2
- In the conclusion section, the limitations of this study, suggested improvements of this work and future directions should be highlighted.
Author Response
Answers for Reviewer 3
The manuscript presents spatial and vertical variability of water quality in a polymictic Polish lake, which is interesting. The subject addressed is within the scope of the journal.
However, the manuscript, in its present form, contains several weaknesses. Appropriate revisions to the following points should be undertaken in order to justify recommendation for publication.
- Full names should be shown for all abbreviations in their first occurrence in texts. For example, GPS in p.3, SD in p.5, etc.
The abbreviations were explained in the paper.
- For readers to quickly catch your contribution, it would be better to highlight major difficulties and challenges, and your original achievements to overcome them, in a clearer way in abstract and introduction.
Introduction section has been re-written according to the reviewers suggestion.
- It is shown in the reference list that the authors have a pertinent publication in this field. This raises some concerns regarding the potential overlap with their previous works. The authors should explicitly state the novel contribution of this work, the similarities, and the differences of this work with their previous publications.
Both papers concern polymictic lake Bikcze. However, the aim, methods, results and discussion are completely different. There is reference to the previous paper only in the discussion part stressed that like Bikcze is susceptible to wind influence and that Chlorophyll concentration is determined by in-lake processes rather than hydrological ones.
- 1 - electrical conductivity, pH, dissolved oxygen, and chlorophyll a are adopted in the analysis. What are the other feasible alternatives? What are the advantages of adopting these parameters over others in this case? How will this affect the results? More details should be furnished.
In-situ measurements of water quality parameters are highly accurate and easy to obtain. Selected parameters were chosen because they represent the physical, chemical as well as biological characteristics of the water. The advantage of the presented method is the possibility of using every measurable parameter. Alternative parameters, e.g. macroelements, nutrients, redox potential, depend (to a various extent) on the basic qualitative characteristics that were subject to our analysis. The use of other parameters would result in a different, unpredictable spatial and vertical distribution.
- 1 - two approaches are adopted to obtain final raster images. What are the other feasible alternatives? What are the advantages of adopting these approaches over others in this case? How will this affect the results? More details should be furnished.
Paper presents two approaches, from top to bottom and nearest neighbor, that optimally present variability of water quality parameters. Alternative could be bottom-top. In this case nearest neighbour method should also be calculated from the bottom (highest depth). Majority of water quality analyses are based on classical approach from the surface layer, to the bottom. Application of bottom to top approach would make alalysis of the results very difficult and comparison to other lakes rather impossible.
- 2 - a hypothesis that homogeneous physicochemical conditions occur in the lake basin is adopted in this study. What are other feasible alternatives? What are the advantages of adopting this hypothesis over others in this case? How will this affect the results? The authors should provide more details on this.
Shallow polymictic, actively mixed lakes are characterized by a stable distribution of water quality parameters in the lake basin, which has been confirmed by numerous studies of lakes in various climatic zones. In addition, the selected spring sampling period favored the mixing of water in the lake. Therefore, it was decided to test the hypothesis about the homogeneity of the physicochemical conditions of water. Dense vertical and horizontal sampling guaranteed objective results. In this context, a relative (nearest neighbor), absent in the literature, methodological solution was used. An alternative hypothesis could be assumption that some parameters vary while others are stabilized spatially and vertically, or assumption of complete variability of water quality paramemters. Regardless of the hypothesis, it would not have any impact on the obtained results.
- 2 - Lake Bikcze is adopted as the case study. What are other feasible alternatives? What are the advantages of adopting this case study over others in this case? How will this affect the results? The authors should provide more details on this.
Lake Bikcze was chosen as the research object for several reasons. The shallow, flat-bottomed lake basin is susceptible to wind and water mixing. The lake bas been anthropogenically cut off from the surface catchment, and the water distribution is easy to identify (as opposed to other polymictic Łęczna-Włodawa lakes). The mixing process is enhanced by meteorological conditions and streams input-dreinage processes. The advantage of the presented method is the possibility to choose any lake as a study object. The choice of Lake Bikcze guaranteed a good example of a 'case study' because the influence of, for example, catchment factors on shaping the water quality in the basin was eliminated.
- 2 - historical records of 2017 are taken. Why are more recent data not included in the study? Is there any difficulty in obtaining more recent data? Are there any changes to situation in recent years? What are its effects on the result?
The data has been obtained in 2017 within the project of National Science Centre. The lake was not anthropogenically transformed at that time. The use of the method in such conditions guarantees its objectivity. The weather conditions ensured the study of functioning of the polymictic lake in the situation of full and intensive mixing of waters
- 3 - inverse distance weight interpolation method is adopted to generate parameter distribution maps. What are other feasible alternatives? What are the advantages of adopting this method over others in this case? How will this affect the results? The authors should provide more details on this.
For any spatial estimation IDW often performs better than other spatial interpolation techniques. The reason lies behind the way it models spatial variability. Kriging and even thin plate spline methods perform smoothing while modelling spatial variability whereas IDW models it by fitting straight lines through the observations. Therefore the estimated values are more similar close to the observed location and get decreased by the distance weights as the estimated locations get further away.
- 4 - Pearson correlation is adopted for the values from the maps. What are other feasible alternatives? What are the advantages of adopting this approach over others in this case? How will this affect the results? The authors should provide more details on this.
Korelacja Pearsona jest metodą powszechnie stosowaną w badaniach środowiskowych, dającą duże możliwości interpretacyjne I porównawcze. Zastosowanie związku korelacyjnego, w odróżnieniu np. od związków allometrycznych, analizy wariancji, itp. Gwarantuje łatwość powtarzalności. Poprawnie dobrana metoda statystyczna, gwarantuje potwierdzenie istniejących zależności pomiędzy zmiennymi.
- 8-9 - “…In turn, the highest variability was shown by differences in average chlorophyll, that is, from −0.09 to 0.04 μg L-1. The smallest variation in the results of average differences was found in pH and DO.…” Some justification should be furnished on this issue.
The presented sentence (from the results section) has been extended in the discussion section
- 10 - “…In the southern part of the lake basin, positive values of chl-a differences indicated a higher concentration of the parameter in the upper layers of the water column. This was probably related to the.…” More justification should be furnished on this issue.
More justification has been provided in MS
- Some assumptions are stated in various sections. Justifications should be provided on these assumptions. Evaluation on how they will affect the results should be made.
The suggested statements have been reviewed.
- The discussion section in the present form is relatively weak and should be strengthened with more details and justifications.
The section has been corrected
- Moreover, the manuscript could be substantially improved by relying and citing more on recent literatures about contemporary real-life water quality case studies such as the followings. Discussions about result comparison and/or incorporation of those concepts in your works are encouraged:
The paper has been completed with more current references, according to the Rewievers suggestion. Specific references pointed by the Reviewer were included in the text.
- Setshedi, K.J., et al., “The Use of Artificial Neural Networks to Predict the Physicochemical Characteristics of Water Quality in Three District Municipalities, Eastern Cape Province, South Africa,” International Journal of Environmental Research and Public Health 18 (10): 5248 2021.
- Deng, T.A., et al., “Machine Learning Based Marine Water Quality Prediction for Coastal Hydro-environment Management,” Journal of Environmental Management 284: 112051 2021.
- Wang, B.B., et al., “Improved water pollution index for determining spatiotemporal water quality dynamics: Case study in the Erdao Songhua River Basin, China,” Ecological Indicators 129: 107931 2021.
- Some inconsistencies and minor errors that needed attention are:
- Replace “…were to a) asses spatial variability…” with “…were to a) assess spatial variability…” in line 60 of p.2
The problematic sentence has been corrected.
- In the conclusion section, the limitations of this study, suggested improvements of this work and future directions should be highlighted.
The section has been modified according to the reviewers suggestion.

Round 2
Reviewer 1 Report
The explanation givem make sense.
Author Response
We would like to thank the Reviewer for the time and effort put into improving our paper. All of the remarks have been taken into considerations. Changes made in the MS according to the Reviewer’s remarks improved the paper tremendously.
Reviewer 3 Report
The most significant comments in the last review (including novelty, major difficulties and challenges, your original achievements to overcome them, etc.) have not been demonstrated satisfactorily. Appropriate revisions to the points raised in the last review should be undertaken in order to justify recommendation for publication.
Author Response
We would like to thank the Reviewer for the time and effort put into improving our paper. All of the remarks have been taken into considerations. All of the raised problems, including novelty, major difficulties and challenges, our original achievements to overcome them, have been addressed in abstract, introduction and conclusion sections. We believe that changes made in the MS according to the Reviewer’s remarks improved the paper tremendously.